# Electron-Induced Repair of 2′-Deoxyribose Sugar Radicals in DNA: A Density Functional Theory (DFT) Study

**DOI:** 10.3390/ijms22041736

**Published:** 2021-02-09

**Authors:** Michael Bell, Anil Kumar, Michael D. Sevilla

**Affiliations:** Department of Chemistry, Oakland University, Rochester, MI 48309, USA; mbell@oakland.edu (M.B.); kumar@oakland.edu (A.K.)

**Keywords:** sugar radical, electron affinity, redox potential, 5′,8-cyclo-guanine, Boltzmann population

## Abstract

In this work, we used ωB97XD density functional and 6-31++G** basis set to study the structure, electron affinity, populations via Boltzmann distribution, and one-electron reduction potentials (E°) of 2′-deoxyribose sugar radicals in aqueous phase by considering 2′-deoxyguanosine and 2′-deoxythymidine as a model of DNA. The calculation predicted the relative stability of sugar radicals in the order C4′^•^ > C1′^•^ > C5′^•^ > C3′^•^ > C2′^•^. The Boltzmann distribution populations based on the relative stability of the sugar radicals were not those found for ionizing radiation or OH-radical attack and are good evidence the kinetic mechanisms of the processes drive the products formed. The adiabatic electron affinities of these sugar radicals were in the range 2.6–3.3 eV which is higher than the canonical DNA bases. The sugar radicals reduction potentials (E°) without protonation (−1.8 to −1.2 V) were also significantly higher than the bases. Thus the sugar radicals will be far more readily reduced by solvated electrons than the DNA bases. In the aqueous phase, these one-electron reduced sugar radicals (anions) are protonated from solvent and thus are efficiently repaired via the “electron-induced proton transfer mechanism”. The calculation shows that, in comparison to efficient repair of sugar radicals by the electron-induced proton transfer mechanism, the repair of the cyclopurine lesion, 5′,8-cyclo-2′-dG, would involve a substantial barrier.

## 1. Introduction

Cellular DNA is damaged by ionizing radiation [1,2,3,4,5,6,7,8,9,10,11] as well as by reactive oxygen species (ROS) [12,13] such as O_2_, ^1^O_2_, O_2_^•−^, OH^•^, NO^•^, NO_2_^•^, ONOO^−^, H_2_O_2_, and peroxy radicals (ROO^•^). Radiation and ROS act synergistically to cause far more damage than each separately [10]. Initially, radiation randomly ionizes each building block of DNA (bases, deoxyribose, and phosphate) and the surrounding water molecules randomly to produce highly reactive ion radicals [2,4,5,6,7,8,9,10,11]. The study of the mechanisms of formation and subsequent reaction of these transient ion radicals is of fundamental importance to the understanding of the extent of DNA damage and related consequences. As an example, irradiations of DNA by a high-energy argon ion-beam (high linear-energy-transfer (LET) radiation) and *γ*-irradiation (a low LET radiation) both produced significant fractions of sugar radicals with the ion-beam irradiated DNA showing a greater fraction than *γ*-irradiation [14]. Since these sugar radicals were formed predominantly along the ion track, where excitations and ionizations are in proximity, it was proposed that base cation radicals in excited states could be the direct precursors of the neutral sugar radicals [14,15]. This hypothesis was extensively tested by Sevilla and his coworkers to produce neutral sugar radicals by exciting the cation radicals of guanine (G^•+^) and adenine (A^•+^) in model systems of nucleosides, nucleotides, and DNA and RNA oligomers using UV-visible light [16,17,18,19,20,21]. The radicals were further characterized by the electron spin resonance (ESR) experiment as C1′^•^, C3′^•^, and C5′^•^ sugar radicals [16,17,18,19,20,21,22,23]. These experimental findings were also supported by the excited state calculations of radical cations of several deoxyribonucleosides and single-stranded dinucleosides [18,22,23,24]. Using EPR (electron paramagnetic resonance)/ENDOR (electron nuclear double resonance), sugar radicals at each of the carbon sites has been shown to result from direct radiation of nucleosides and nucleotides in the solid state [25,26]. The experimental formation of the C5′^•^ sugar radical from 2′-deoxyguanosine radical cation (2′-dG^•+^) was proposed to occur via a proton-coupled hole transfer (PCHT) mechanism from the base to the sugar site which was supported by the DFT [27].

The indirect effect of radiation on water-containing systems results in formation of hydroxyl radicals (OH^•^) from ionization of water followed by rapid deprotonation. Waters surrounding the DNA ionizations of the first solvation shell (ca. 9 waters) undergo hole transfer to the DNA while the second layer of waters form OH^•^ [28]. OH^•^ is highly reactive and found to be a highly damaging entity as it accounts for ca. half of radiation damage to cellular DNA by low LET radiation [1,10,29]. The dominant reaction of OH^•^ with DNA/RNA bases is the addition reaction (k ≥ 10 ^9^ M^−1^ s^−1^ [29]) at C4, C5, and C8 sites of guanine and adenine and C5 and C6 sites of thymine, cytosine, and uracil [1,29,30,31,32,33,34,35,36,37]. OH^•^ reacts with sugar moiety in DNA via the hydrogen abstraction reaction from carbon sites of the sugar ring to produce C1′^•^, C2′^•^, C3′^•^, C4′^•^, and C5′^•^ neutral sugar radicals [29,38,39,40,41,42]. Balasubramanian et al. [38] showed that the rate of hydrogen abstraction by OH^•^ from different sites of deoxyribose in DNA depends on the exposure of the sugar hydrogen atoms to the solvent and lies in the order H5′ > H4′ > H3′ ≈ H2′ ≈ H1′. We note that Bernhard and coworkers [43,44] and Greenberg and coworkers [45,46] observed the formation of C1′^•^ predominantly in *γ*-irradiated DNA but this formation may be from hole transfer from the ionized DNA bases and not a result of OH^•^ attack [22].

Formation of neutral sugar radicals in DNA/RNA is known to cause strand breaks [47,48,49], base release [50], cross-link formation [51,52,53,54], and mutagenesis [41,55,56]. There are several studies which indicated that sugar radicals in DNA/RNA are the locus for further oxidation by several chemical species [1,30,41,57,58] which also leads to DNA damage via strand breaks. Regarding the one-electron reduction of sugar radicals, Razskazovskii et al. [59] proposed the reductive repair mechanism of C1′^•^ in DNA by cysteamine. Such thiol repair processes are well known and have been established as a major component of the oxygen enhancement effect of radiation as repair by thiols is hindered in the presence of oxygen. Knowledge about the reductive properties of sugar radicals in DNA is important to the understanding of the repair mechanism of sugar radicals [60].

In this study, we calculated the structure, electron affinity, and standard one-electron reduction potentials (E°) of C1′^•^, C2′^•^, C3′^•^, C4′^•^, and C5′^•^ neutral sugar radicals in 2′-deoxyguanosine (2′-dG) and 2′-deoxythymidine (2′-dT). These nucleosides are considered as models for the base and sugar components of DNA. These nucleosides were chosen because guanine has the lowest and thymine has the highest one-electron reduction potential of the four DNA bases. [61,62]. Experimentally, it is found that C5′^•^ of 2′-dG undergoes a cyclization reaction with the C8 atom of the guanine moiety and produces two diastereoisomeric forms: (i) 5′(R),8-cyclo-2′-deoxyguanosine-7-yl radical and (ii) 5′(S),8-cyclo-2′-deoxyguanosine-7-yl radical [11,30,52,63,64,65]. The structure of sugar radicals and 5′,8-cyclic guanine considered in this work is shown in Scheme 1. Our calculation shows that all sugar radicals in Scheme 1 are efficiently reduced by the solvated electron and possess higher electron affinities than pyrimidines. Further, to test the proposed repair mechanism of Razskazovskii et al. [59], we explored the reaction of one-electron reduced sugar radicals with water and found that these reduced sugar radicals are protonated from solvent. Based on our calculations, we propose that electron-induced repair of sugar radicals in DNA would be in competition with chemical repair by thiols, or further reaction of the sugar radicals to form strand breaks and associated base release.

## 2. Methods of Calculation

In this work, ωB97XD density functional along with 6-31++G** basis set was used to calculate the optimized structures of neutral and one-electron reduced sugar radicals of 2′-deoxyguanosine and 2′-deoxythymidine. ωB97XD is a long-range corrected hybrid density functional with damped atom–atom dispersion corrections developed by Chai and Head-Gordon [66,67]. ωB97XD and its variants is a good choice to satisfactorily describe the ground and excited state electronic properties, ionization energies, electron affinity, and redox potentials of nucleic acid bases and base pairs in our recent publications [58,61,68,69,70,71,72,73] and others [74,75]. The use of 6-31++G** basis set is found suitable to produce molecular properties [58,61,68,69,70,71,72,73]. Vibrational frequencies were calculated to ensure that optimized structure corresponds to local minimum on the potential energy surface.

The standard one-electron reduction potential E° vs. SHE of neutral sugar radicals of 2′-dG and 2′-dT were calculated as
(1)$$E^{o}=\frac{-\Delta G_{\mathrm{sol}}^{o}}{\mathrm{nF}}-\mathrm{SHE}$$
where SHE is the absolute standard hydrogen electrode potential and F is the Faraday constant. The reported values of the reference value of the absolute potential of the SHE in water vary in the range 4.28 V–4.44 V [76,77]. In this work, we used the IUPAC (Inter-national Union of Pure and Applied Chemistry) recommended value 4.44 V for SHE [77]. The Faraday constant F is equal to 23.061 kcal·mol^−1^·V^−1^ (1 eV V^−1^) [8]. In Equation (1) *n* = 1 since standard reduction reaction of neutral sugar radicals in this study is a one electron process. E° was calculated using the direct method instead of considering the complete thermodynamic cycle because the results by both the methods are comparable [78]. The solvation free energy in aqueous phase ($-\Delta G_{\mathrm{sol}}^{o}$) in Equation (1) was calculated from one-electron reduction reaction Equation (2)
(2)$$S_{\mathrm{sol}}^{•}+e_{g\;}^{-}\;\overset{\Delta G_{\mathrm{sol}}^{o}}{\to }{\;S}_{\mathrm{sol}}^{-}$$
$\Delta G_{\mathrm{sol}}^{o}=G^{o}\left(S_{\mathrm{sol}}^{-}\right)-G^{o}\left(S_{\mathrm{sol}}^{•}\right)-G^{o}\left(e_{g}^{-}\right)$. In Equation (2) the free energy of the gas phase electron, G° (e^−^_g_), of −0.867 kcal/mol (−0.04 eV) was obtained from Fermi–Dirac statistics [79]. Gibbs free energy (G°) was calculated at 298 K and 1 atm from vibrational analysis of the molecule in question. All calculations were carried out in the aqueous phase (ε = 78.4) via the integral equation formalism variant polarized continuum model (IEF-PCM) solvation model of Tomasi et al. [80]. The complete methodology in this work is abbreviated as ωB97XD-PCM/6-31++G**. Calculations were done using the Gaussian 16 suite of programs [81] and spin density, and molecular structures were plotted using the GaussView [82] and Jmol [83] molecular modeling softwares. The calculated electronic and free energies (G) of species considered in this work are given in the Appendix A.

## 3. Results and Discussion

### 3.1. Structures and Populations

#### 3.1.1. Sugar Radicals

In their optimized structures, each carbon atom (C1′, C2′, C3′, and C4′) of sugar moiety (deoxyribose) and C5′ in ribonucleoside and deoxyribonucleoside adopts a near tetrahedral conformation [84,85], however, on radical formation by hydrogen abstraction, the corresponding carbon on which the radical is centered undergoes a large structural change [58,84,85]. The ωB97XD-PCM/6-31++G** optimized geometries of neutral sugar radicals (C1′^•^, C2′^•^, C3′^•^, C4′^•^, and C5′^•^) of 2′-dG and 2′-dT show that the sugar moiety in both the cases has similar conformation and the nature of the base (G or T) attached at C1′ of sugar ring has virtually no effect, see Scheme 1. To represent the degree of planarity of the radical site on deoxyribose ring, we sum three angles centering the carbon on which the radical is centered, for an ideal tetrahedral conformation such as CH_4_ the sum of three angles centering carbon is 327° and for complete planar conformation the corresponding sum should be 360°. The ωB97XD-PCM/6-31++G** optimized structure of C1′^•^ of 2′-dG shows a large structural change at C1′ center. The C1′ center which was non-planar, the sum of the three angles centering C1′ (N9−C1′−O + O−C1′−C2′ + C2′−C1′−N9) is ca. 329° before radical formation becomes quite planar and the sum of corresponding angles is ca. 347°. C2′^•^ of 2′-dG becomes near planar having the sum of angles centering the C2′ atom ca. 359°. For C3′^•^, C4′^•^, and C5′^•^ of 2′-dG the sum of angles centering C3′, C4′, and C5′ atoms is 345°, 357°, and 355°, respectively. Thus, C2′, C4′, and C5′ radicals become near planar, while C1′ and C3′ radicals have significant non-planarity. Similarly, the optimized C1′^•^, C2′^•^, C3′^•^, C4′^•^, and C5′^•^ of 2′-dT has the angles as 349°, 359°, 347°, 353°, and 355°, respectively. These results are in close agreement with earlier studies [58,84,85]. The relative free energies (ΔG (stabilities)) of C1′^•^, C2′^•^, C3′^•^, C4′^•^, and C5′^•^ of 2′-dG and 2′-dT calculated at 298 K are presented in Table 1 and Table 2, respectively. From Table 1, it is evident that relative stabilities of sugar radicals of 2′-dG and 2′-dT are in the order C4′^•^ > C1′^•^ > C5′^•^ > C3′^•^ > C2′^•^. The relative stability between C4′^•^ and C1′^•^ are comparable (ca. 1.5 kJ/mol, see Table 2) and thus both can form at room temperature. There are few calculations which predicted the relative stabilities of sugar radicals in the order C1′^•^ > C4′^•^ > C5′^•^ > C3′^•^ > C2′^•^ [84,85,86,87]. The calculated Boltzmann populations of C1′^•^, C2′^•^, C3′^•^, C4′^•^, and C5′^•^ of 2′-dG and 2′-dT at 298 K show that the thermodynamic stability, based on Boltzman equilibrium weighting, of the formation of C4′^•^ and C1′^•^ is very high ca. 57% and ca. 36% with a small fraction of C5′^•^ (6%), see Table 1 and Table 2, respectively. We note that radiation-produced sugar radicals from direct ionization followed by deprotonation, do not follow this equilibrium distribution and show no formation of C4′^•^ or C2′^•^ with significant yields of C1′^•^, C3′^•^, and C5′^•^ in the order C5′^•^ > C3′^•^ > C1′^•^ [17], see Table 3. This is attributed to the distribution of the hole on the sugar ring with sites with highest hole density being favored for deprotonation [17]. In addition, OH^•^ attack in DNA systems is at the base (80%) and the sugar sites (20%). The high reactivity of the OH^•^ toward abstraction would make it nonselective, however, its reaction is modified by kinetic factors such as accessibility of the C–H sugar sites in the DNA structure. Thus, the OH^•^ attack also shows sugar radical yields not in accord with the equilibrium distribution with C5′^•^ > C4′^•^ > C3′^•^ > C2′^•^ > C1′^•^ [38], see Table 3.

In Table 3 the yields by OH^•^ attack on the sugar sites in DNA in solution are compared to the overall solvent accessibility to the sites. The comparison shows that attack of the OH^•^ is limited by accessibility factors rather than the bond strength [38]. The high rates of hydrogen abstraction by OH^•^ for all sugar sites make relative bond strengths less significant than accessibility.

#### 3.1.2. One-Electron Reduced Sugar Radicals (Anions)

Free and solvated electrons formed in irradiated DNA systems may recombine with sugar radicals or add to the bases. Thus, an understanding of the VEA (vertical electron affinity), AEA (adiabatic electron affinity), and E° (standard reduction potential) for electron addition is important to understanding the electron repair process. The one-electron reduction of neutral sugar radicals (C1′^•^, C2′^•^, C3′^•^, C4′^•^, and C5′^•^) of 2′-dG and 2′-dT produces sugar anions (C1′^−^, C2′^−^, C3′^−^, C4′^−^, and C5′^−^) of 2′-dG and 2′-dT. The ωB97XD-PCM/6-31++G**-calculated optimized structures of sugar anions (C1′^−^, C2′^−^, C3′^−^, C4′^−^, and C5′^−^) of 2′-dG and 2′-dT show that large structural changes occur mainly in the sugar moiety. From the optimized structures of sugar anions of 2′-dG and 2′-dT, it is found that sugar anions become quite non-planar in comparison to their radical structure which was found to be planar. The C1′ anion of 2′-dG becomes quite non-planar and the sum of the three angles (N9−C1′−O + O−C1′−C2′ + C2′−C1′−N9) centering C1′ is 317°, however, for C1′^•^ the corresponding angle is 347°, see discussion in Section 3.1.1. For C2′, C3′, C4′, and C5′ anions of 2′-dG, the sum of three angles centering the corresponding carbon atom is 334°, 310°, 328°, and 318°, respectively. Similarly for C1′^−^, C2′^−^, C3′^−^, C4′^−^, and C5′^−^ of 2′-dT the angles is 316°, 337°, 311°, 327°, and 323°, respectively. Very interestingly, the C2′ radical which has the lowest stability among the other C1′, C3′, C4′, and C5′ radicals (see Table 1 and Table 2) becomes most stable in the reduced state. The relative free energy (stability) of the sugar anions of 2′-dG and 2′-dT follows the order C2′^−^ > C4′^−^ > C1′^−^ > C3′^−^ > C5′^−^ and C2′^−^ > C4′^−^ > C5′^−^ > C3′^−^ > C1′^−^, respectively.

### 3.2. Spin Density Distributions of Sugar Radicals and HOMO of Their Anions

The ωB97XD-PCM/6-31++G**-calculated spin density plots of sugar radicals of 2′-dG and highest occupied molecular orbital (HOMO) of their corresponding anions are presented in Figure 1 and of those of 2′-dT are presented in the Appendix A. For a radical species (odd electron system) the spin density distributions presented as a 3-dimensional visual plot characterize the nature of odd electron distributions [88] within the molecule. The anion, produced by one-electron reduction of a radical, is an even electron system and HOMO plot of anion gives the visual information about the perturbation induced to the radical after an excess electron attachment. From the spin density plot of 2′-dG(C1′^•^), we infer that spin density is mostly localized (>90%) on the C1′ atom with a small fraction on N7, C8, and C4 atoms of guanine and on the O atom of the sugar moiety. The spin distribution is π in nature as this site acquires planarity on radical formation. The spin density plots of C2′^•^, C3′^•^, C4′^•^, and C5′^•^ of 2′-dG also show that the spin is well localized on the corresponding carbon radical site of the sugar ring and spin distributions in all these radicals are π in nature, see Figure 1. The HOMO plot of 2′-dG(C1′^−^) shows similar electron distributions as obtained for the spin density plot of 2′-dG(C1′^•^) which clearly shows that an excess electron attaches to the same half-filled MO of the 2′-dG(C1′^•^). From the visual inspection of the HOMO of 2′-dG(C1′^−^), it is also noticed that the electron distribution in the anion, 2′-dG(C1′^−^), is σ in nature pointing outward with respect to the plane constituting the C_1′_ atom as a center. This is obvious as the C1′ atom in 2′-dG(C1′^−^) becomes non-planar while in 2′-dG(C1′^•^) it is planar, see Figure 1. The HOMOs of 2′-dG(C2′^−^)-2′-dG(C5′^−^) also show a feature similar to that discussed for 2′-dG(C1′^−^) and they are also σ-type, see Figure 1. Similar results were also found for 2′-dT, see Appendix A.

### 3.3. Electron Affinity and Reduction Potential (E°) of Sugar Radicals

The ωB97XD-PCM/6-31++G**-calculated vertical (VEA) and adiabatic (AEA) electron affinities of sugar radicals of 2′-dG and 2′-dT are presented in Table 1 and Table 2, respectively. The VEA of sugar radicals (C1′^•^, C2′^•^, C3′^•^, C4′^•^, and C5′^•^) of 2′-dG and 2′-dT are quite similar (maximum difference of 0.1 eV) and range from 1.6 eV to 2.8 eV, respectively, see Table 1 and Table 2. The average AEA of these sugar radicals are substantially higher and range from 2.7 eV to 3.2 eV. C2′^•^ has the highest AEA (3.2 eV) while C1′^•^ has the lowest AEA (2.6 eV). It is noted that in DNA, pyrimidines are the prime site for electron attachment and the calculated AEA of thymine and cytosine in the aqueous phase ranges from ca. 1.6 to 2.2 eV [89,90,91,92], respectively. Thus, sugar radicals have far higher electron affinities (>0.4 eV) than the bases and would be a locus for an excess electron attachment in DNA. From the HOMO plots of sugar anions (see Figure 1 and Appendix A), it is evident that the molecular orbital, HOMO, is mainly confined on the original radical site in the sugar moiety.

The ωB97XD-PCM/6-31++G**-calculated standard one-electron reduction potentials (E°) of C1′^•^, C2′^•^, C3′^•^, C4′^•^, and C5′^•^ of 2′-dG and 2′-dT, presented in Table 1 and Table 2, are similar. The average E° of C1′^•^, C2′^•^, C3′^•^, C4′^•^, and C5′^•^ is −1.84, −1.25, −1.73, −1.72 and −1.79 V, respectively. The experimental E° of solvated electron (e^−^_aq_) is −2.87 V [69,76,77,93,94] and E° of DNA/RNA bases measured in DMF (*N*,*N*-dimethylformamide) by Seidel et al. [62] are adenine (−2.52 V), guanine (<−2.76 V), cytosine (−2.35 V), thymine (−2.18 V), and uracil (−2.07 V) and these values were satisfactorily reproduced using the G4 level of theory by Kumar et al. [61]. In going from DMF to aqueous phase, E° needs a very marginal correction of solvation energy of −0.02 eV; however, protonation from the surrounding aqueous medium will occur and the resultant E° will be reduced (see Section 4). From a comparison of the E° of sugar radicals with respect to e^−^_aq_ and DNA/RNA bases, it is evident that sugar radicals have a far higher probability to be reduced by the e^−^_aq_ than the bases.

### 3.4. Cyclization of C5′^•^ and C8 of Guanine

The C5′^•^ of 2′-dG undergoes a cyclic reaction in which the C5′^•^ attacks the C8 of guanine and produces the unique cyclic sugar−base adduct radicals as shown in Scheme 1. These cyclic adducts are present in two diastereomeric forms and have been identified in the mammalian cellular DNA in vivo [11,30,52,63,64,65]. The two isomers of the radicals are: (i) 5′(R),8-cyclo-2′-deoxyguanosine-7-yl radical and (ii) 5′(S),8-cyclo-2′-deoxyguanosine-7-yl radical and the rate of cyclization reaction is ca. 1 × 10^6^ s^−1^ [30]. The structures of these two isomers (shown in Scheme 1) in their radical and one-electron reduced (anion) form were optimized using the ωB97XD-PCM/6-31++G** method. The optimized structures of these two isomers show that the sugar ring of these cyclic isomers adopts the O-exo conformation as found experimentally using NMR spectroscopy [63] and theory [58,95,96]. The spin density distribution plot and HOMO of the radical and anion of these two isomers are shown in Figure 2. From Figure 2, it is inferred that spin density in these two cyclic radical isomers is localized on guanine. About 50% of total spin is localized on N7 and rest is distributed on the other atoms of guanine. Thus, on cyclization, spin is transferred from C5′ to guanine in these two cyclic isomers and the nature of spin distribution is π-type. The HOMO of both the anions is similar to their corresponding spin density plots and also localized on guanine.

In Table 4, we present the ωB97XD-PCM/6-31++G**-calculated relative free energy (stability), population, electron affinity, and E° of 5′(R),8-cyclo-2′-deoxyguanosine-7-yl and 5′(S) and 8-cyclo-2′-deoxyguanosine-7-yl radicals. From Table 1, we see that the 5′(R),8-cyclo-2′-deoxyguanosine-7-yl radical is slightly more stable than the 5′(S),8-cyclo-2′-deoxyguanosine-7-yl radical by 1.44 kJ/mol. Also, we note that 5′(R) and 5′(S) radical isomers are more stable than their precursor (2′-dG(C5′^•^)) by 6.3 kcal/mol and 6 kcal/mol and 5′(R) and 5′(S) anions are more stable than 2′-dG(C5′^−^) by 11.7 kcal/mol and 17.4 kcal/mol, respectively. The calculated Boltzmann population at 298 K shows that both 5′(R) and 5′(S) radicals will be present as 64% and 36%. The calculated VEA, AEA, and E° of 5′(R) and 5′(S) radicals are comparable to those of C1′^•^, C2′^•^, C3′^•^, C4′^•^ and C5′^•^ of 2′-dG and 2′-dT, see Table 1, Table 2 and Table 3. Thus, like sugar radicals, these isomers (5′(R) and 5′(S)) are also efficiently reduced by the solvated electron in comparison to DNA bases in DNA.

In the present context, it is pertinent to discuss the relative stability of (i) neutral (diamagnetic product) 5′(R),8-cyclo-2′-deoxyguanosine and (ii) 5′(S),8-cyclo-2′-deoxyguanosine, see structures in the Appendix A. Dizdaroglu and coworkers [65] reported the yields (number of lesions per Gy (10^6^ DNA bases)^−1^) of (i) 5′(R),8-cyclo-2′-deoxyguanosine and (ii) 5′(S),8-cyclo-2′-deoxyguanosine in mammalian DNA using LC/MS and GC/MS. They estimated the yield of 5′(R),8-cyclo-2′-deoxyguanosine as 0.80 ± 0.03 Gy^−1^(10^6^ DNA bases)^−1^ (average of two methods) and for 5′(S),8-cyclo-2′-deoxyguanosine the average yield was 2.65 ± 0.03 Gy^−1^(10^6^ DNA bases)^−1^ ratio of the 5′(R)/5′(S) is ca. 0.3. We also, optimized these structures in the neutral state and calculated the free energy of both the conformations at 298 K using the ωB97XD-PCM/6-31++G** method, see Appendix A. Our calculated Boltzmann populations for 5′(R),8-cyclo-2′-deoxyguanosine and for 5′(S),8-cyclo-2′-deoxyguanosine are 28.4% and 71.6%, respectively, and calculated 5′(R)/5′(S) ratio is 0.4 which is in excellent agreement with experimental quantification of 5′(R) and 5′(S) isomers by Dizdaroglu and coworkers [65]. However, for the radical the ratio of 5′(R)/5′(S) is 1.79, see Table 4 and needs to be verified experimentally.

## 4. Protonation of Reduced Sugar Radicals from Water

The one-electron reduced sugar radical species described above would be rapidly protonated in by nearby waters for any DNA system with available waters of hydration. To model the proton transfer (PT) reaction from water to one-electron reduced sugar radicals, we placed three discrete water molecules around the anionic carbon site in question and fully optimized the structure using the ωB97XD-PCM/6-31++G** method. The PCM was employed to model the remaining solvent and account for solvation of ions. We considered three waters to make the calculations feasible instead of considering the full solvation shell which needs enormous CPU time and not possible given our present computational resources. In our approach, we considered sugar anions (C1′^−^, C2′^−^, C3′^−^, C4′^−^, and C5′^−^) of 2′-dT in the presence of three waters chosen to provide hydrogen bonding stabilization of the resultant OH anion. From the optimized structures of 2′-dT(C1′^−^) to 2′-dT(C5′^−^) with 3H_2_O, we found that all the sugar anions of 2′-dT were protonated from water during geometry optimization without any barrier except [2′-dT(C1′^−^) + 3H_2_O]. Further, protonation of C1′ anion as well as all other sugar anion is exothermic. The optimized structure of 2′-dT(C1′^−^) + 3H_2_O, before and after PT is depicted in Figure 3. From Figure 3, it is evident that the reaction is significantly exothermic by the 1.45 eV calculated from the free energies of product and reactant. The HOMO which was localized mainly on C1′ before PT is localized on the OH^−^ (hydroxide anion) in water after PT. The ωB97XD-PCM/6-31++G**-calculated Mulliken charges show that ca. −0.8 e (in the unit of magnitude of the electronic charge) stays on the OH moiety while −0.2 e resides on the 2′-dT. The localization of HOMO on OH^−^ shows that in this system, (2′-dT + OH^−^) OH^−^ has the lower ionization potential. Adriaanse et al. estimated the adiabatic ionization potential (AIP) of aqueous OH^−^ as 6.1 eV [97] which is lower than the AIP of thymine (6.4 eV) and deoxyribose (7.43 eV) calculated using theory [98]. The HOMOs of (C2′^−^, C3′^−^, C4′^−^, and C5′^−^) of 2′-dT + 3H_2_O after PT is shown in Appendix A. Experimentally, it is found that in the presence of a physiological concentration of thiol (GSH 2.5 mM), 2′-dT(C5′^•^) is completely repaired as 2′-dT [30,99]. However, C5′^•^ of 2′-dG is partially repaired and in competition to form 5′,8-cyclo-2′-dG [30,99]. Thus, our calculation clearly shows that the lesion produced on deoxyribose of nucleosides/tides in the form of sugar radical can be efficiently repaired by an “electron-induced PT reaction mechanism” as proposed by Razskazovskii et al. [59]. Our calculation also points out that repair of cyclic lesion 5′,8-cyclo-2′-dG will involve a high barrier of more than 12 to 17 kcal/mol to break the C5′−C8 bond since cyclic anions are more stable than 2′-dG(C5′) anion by 12 and 17 kcal/mol. These model systems show the protonation process should be fast and will affect the redox potential for this reason we computed the E° for the process of reduction for the half reaction:(3)$${Cn^{\prime }}^{•}+e^{-}+H^{+}\to Cn^{\prime }H\;(n\;=\;1–5)$$

Since the SHE is given by the reaction:(4)H++ e−→12H2

Equation (3) minus Equation (4) gives Equation (5):(5)Cn′•+12H2→Cn′H

The free energy change found for Equation (5) gives the potential for Equation (3) vs. the SHE, i.e., $E^{o}=-\Delta G^{o}\left(5\right)/\mathrm{RT}$.

These values are given in Table 2 and Table 3 in the last column and show the substantial reduction for the equilibrium thermodynamic value of E° with protonation of the anions except for C2′. In Equations (3) and (5) Cn′H represents the neutral 2′-dG and 2′-dT, respectively. The value of 1/2(H_2_) was calculated using Gaussian 4 (G4) level of theory in PCM [81]. This method eliminates the uncertainties of the proton solvation energy and the SHE when employing the usual methods. We compare the two methods in the Appendix A and find them equivalent to within 0.08 V.

## 5. Conclusions

From this study, we found that neutral sugar radicals C1′^•^, C2′^•^, C3′^•^, C4′^•^, and C5′^•^ of 2′-dG and 2′-dT have AEA in the range ca. 2.6 to 3.2 eV (see Table 1 and Table 2). Since the DNA bases have substantially lower AEAs of 1.6–2.2 eV, sugar radicals will be the greatly preferred sites for excess electron attachment in DNA [89,90,91,92]. C1′^•^ has the lowest AEA (2.6 eV) and C2′^•^ has the highest AEA (3.2 eV). From the spin density plots of sugar radicals, it is evident that more than 90% of spin is localized on carbon radical site of the sugar moiety and the nature of spin distribution is π-type as the carbon radical center in the sugar ring becomes quit planar, see Figure 1 and Appendix A. The anion of these sugar radicals has doubly occupied molecular orbitals with singlet ground state and the HOMO plots of the anions resemble the spin density plots for the radicals as expected. The ωB97XD-PCM/6-31++G**-calculated E° of sugar radical of 2′-dG and 2′-dT, are in the range −1.25 to −1.84 V which is higher than the E° of e^−^_aq_ and DNA/RNA bases. Thus, sugar radicals in DNA are far more probable sites to be reduced by the solvated electron in comparison to the DNA bases.

The relative stability of the various sugar radicals reported in Table 1, Table 2 and Table 3 is not found to be the major determiner for sugar radical yields in DNA in attack by radiation or by highly reactive OH^•^. The yields from ionizing radiation are driven by the hole distribution in the sugar ring [17,18] while the hydroxyl radical attack is controlled by the accessibility of the various sugar C−H bonds to the diffusing OH^•^ [38].

The 2′-dG(C5′^•^) species is experimentally found to undergo cyclization reaction by forming bond between C5′ and C8 of guanine. These are present in two diastereomeric (i) 5′(R),8-cyclo-2′-deoxyguanosine-7-yl^•^ and (ii) 5′(S),8-cyclo-2′-deoxyguanosine-7-yl^•^ forms and populated as 64% (5′(R)) and 36% (5′(S)) as calculated using Boltzmann distribution. These isomers are ca. 6 kcal/mol more stable than their precursor (2′-dG(C5′^•^)), and anions of these cyclic isomers are ca. 12 kcal/stable more stable than the 2′-dG(C5′^−^). These two cyclic radicals also have comparable AEA and E° as calculated for sugar radicals, see Table 1, Table 2 and Table 3, and can be reduced by solvated electron. No experimental results are reported for the relative yields of the radical diastereomers; however, for the diamagnetic molecular products the 5′(R),8-cyclo-2′-deoxyguanosine and 5′(S),8-cyclo-2′-deoxyguanosine, the ratio of the 5′(R)/5′(S) is ca. 0.3 [65]. The DFT-calculated relative energies of the two diasteriomers give thermodynamic populations yielding a value of the 5′(R)/5′(S) ratio of ca. 0.4 which is in good agreement with the experimental ratio, see Appendix A and Appendix A.

The reaction of sugar anions (C2′^−^, C3′^−^, C4′^−^, and C5′^−^) of 2′-dT with 3H_2_O proceeds with a barrierless PT from a water to the anionic site except C1′^−^ may involve some barrier for the PT from water to C1′. Overall, the ωB97XD-PCM/6-31++G**-calculated PT reaction was found to be highly exothermic for example, C1′-protonated anion (2′-dT(C1′^−^ + H^+^) + OH^−^ + 2H_2_O)) was found to be 1.45 eV (ca. 33 kcal/mol) more stable than the C1′ anion (2′-dT(C1′^−^) + 3H_2_O)), see Figure 3. The HOMO which was localized on the C1′ site before PT now localizes on the OH^−^ after proton transfer (Figure 3). In comparison to the efficient repair of sugar radicals by electron-induced proton transfer mechanism, the repair of 5′,8-cyclo-2′-dG involves a substantial barrier. We note it is well established that the cyclopurine lesions such as 5′,8-cyclo-2′-dG, cannot be repaired by the base excision repair pathway and are poorly repaired by the nucleotide excision repair pathway [100]. Their difficulty in repair by electron-induced mechanisms shown in this work only adds to the accumulation of these harmful lesions which can lead to mutations and genomic instability.

## Data Availability

Not applicable.

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
