# Peer review of "Electron-Induced Repair of 2′-Deoxyribose Sugar Radicals in DNA: A Density Functional Theory (DFT) Study"

_ijms, 2021, doi:10.3390/ijms22041736_

Round 1

Reviewer 1 Report

Line 98, the studied systems are relatively small. Please explain why you have not used 6-311++G** basis set?

Equations (1) and (2), please try to improve their quality by writing them in word equations editor instead of doing copy-paste with low resolution. It looks bad now.

Line 117, word “standard” is missing.

Were the initial structures just drawn or taken from crystal structure files?

Line 120, some editorial mistakes here.

Table 1, it would be nice also to present the electronic energies (without ZPVE and thermal corrections) of each calculated species.

Figure 1 splits the text in the middle of the sentence, please correct.

Line 276: “has a very important chemistry” -  this is too colloquial.

Line 342, I don’t say it is wrong, but why this particular number of water molecules (three)? Besides, could you explain how the initial coordinates of water molecules were stated? More precisely, how did you chose the place to put the water molecules (before optimization), based on what?

Equations (3), (4) and (5), please try to improve their quality by writing them in word equations editor instead of doing copy-paste with low resolution. It looks bad now.

Author Response

Reply to the Reviewer’s comments on the manuscript entitled “Electron-Induced Repair of 2'-Deoxyribose Sugar Radicals in DNA: A Density Functional Theory (DFT) Study” by Bell et al.

    We thank the Reviewers for their valuable suggestions/comments which have improved the revised manuscript.    Below we give the reviewers comments and our responses.  We note that the changes made in the manuscript itself are highlighted in yellow.

     Sincerely,

       Michael D. Sevilla

Reviewer 1

Comment 1- Line 98, the studied systems are relatively small. Please explain why you have not used 6-311++G** basis set?

Response- From our earlier calculations, we noticed that 6-31++G** basis set is quite adequate for computing the various properties like ionization potential, electron affinity, and redox potential. These works have been referenced in the manuscript (refs 58,61,68 – 73) . Thus, we used 6-31++G** basis set. We have clarified the revised manuscript by the addition of “ The use of 6-31++G** basis set is found suitable to produce molecular properties [58,61,68–73].” in section 2.

Comment 2- Equations (1) and (2), please try to improve their quality by writing them in word equations editor instead of doing copy-paste with low resolution. It looks bad now.

Response- As suggested by the Reviewer, eqs (1) and (2) have been written using word equation editor.

Comment 3- Line 117, word “standard” is missing.

Response- The word “standard” has been inserted.

Comment 4- Were the initial structures just drawn or taken from crystal structure files?

Response- The initial structures are just drawn in scheme 1.

Comment 5- Line 120, some editorial mistakes here.

Response- This is corrected  “statics” to “statistics”

Comment 6- Table 1, it would be nice also to present the electronic energies (without ZPVE and thermal corrections) of each calculated species.

Response- As per suggestion of the Reviewer, we tabulated the electronic energies (without ZPVE and thermal corrections (free energies)) of each species considered in this work are presented in the supporting information. This is updated in Section 2 as “The calculated electronic (SCF without zero point energy correction) and free energies (G) of species considered in this work are given in the supporting information” and in the foot note (a) of Tables 1 and 2, respectively.

Comment 7- Figure 1 splits the text in the middle of the sentence, please correct.

Response- This was just a minor technical issue and has been corrected in the revised manuscript.

Comment 8- Line 276: “has a very important chemistry” -  this is too colloquial.

Response- We now modified the sentence in revised manuscript as "The C5' radical of 2'-dG undergoes a cyclic reaction in which the C5' attacks the C8 of guanine and produces the unique cyclic sugar-base adduct radicals as shown in scheme 1.”

Comment 9- Line 342, I don’t say it is wrong, but why this particular number of water molecules (three)? Besides, could you explain how the initial coordinates of water molecules were stated? More precisely, how did you chose the place to put the water molecules (before optimization), based on what?

Response- In the anion structures each carbon in question has maximum electron density (see the HOMO and spin density plots), thus in preparing the initial structure we placed three waters around the carbon atom in the hydrogen-bonded configuration approximately more than 2.5 angstrom away from the carbon radical site.  These structures were completely optimized.   We found that for only the C1 anion some barrier for the proton transfer (see Figure 3 in the manuscript) while for other cases proton transfers from water without a barrier. We consider, three waters so that hydroxide ion (OH-) should stabilized by hydrogen bonding from the remaining two waters in addition to PCM.   We have made some changes in wording in this portion for clarity.

Comment 10- Equations (3), (4) and (5), please try to improve their quality by writing them in word equations editor instead of doing copy-paste with low resolution. It looks bad now.

Response- As suggested by the Reviewer, eqs (3) - (5) have been written using word equation editor.

Reviewer 2 Report

Review of the submission “Electron-Induced Repair of 2'-Deoxyribose Sugar Radicals in DNA: A Density Functional Theory (DFT) Study” submitted to the International Journal of Molecular Sciences by Bell, Kumar and Sevilla.

This is an interesting and well written article.  I have found a few typos and I have a few comments that should be easy to fix.

Page 3, line 14.  This should be .. in this study is a one electron process.

Page 3, line 19.  These ΔG’s  are way too big. 

Page 3, line 20.  I think that should be ….Fermi-Dirac statistics.

Page 4, line 40 .  The prime symbols 1ʹ etc here are way too small. nb there are lots of these in this manuscript.

Page 5, line 85.  (see eq 5)?  This is on page 11 and doesn’t seem right?

Page 6, line 23.    …presents in a 3-dimensional visual plot?  I don’t see this plot in reference 88?

Page 6, line 36.  Here again the C1ʹ is way too small.  Same thing on P8 line74.

Page 10, line 46.  …using the free energy?  The free energy of what?

Page 11, line 91.  “resembles with”?  this should be something like plot of anions resembles that of the spin density of radicals.

Page 11, line 95.  “in compare to bases”?  this should be something like …in comparison to the DNA bases”.

Page 12, line 23,  make this …involves a substantial barrier.

Page 12, line 41.  Remove .Gamma. and replace with …. γ-irradiated …

Page 12, line 47.  OF should be of, and And should be and…

Page 12, line 56.  No authors listed.  So add, Dizdaroglu, Lloyd.

Page 14, line 55.  Don ‘t need this † here. 

Page 14, line 64, H. Bul. should be H·and OH·

Page 15, line 73,  Ds should be dS, and Ds should be dS.

Page 15, this should be pKas .

Author Response

Reply to the Reviewer’s comments on the manuscript entitled “Electron-Induced Repair of 2'-Deoxyribose Sugar Radicals in DNA: A Density Functional Theory (DFT) Study” by Bell et al.

    We thank the Reviewers for their valuable suggestions/comments which have improved the revised manuscript.    Below we give the reviewer 2 comments and our responses.  We note that the changes made in the manuscript itself are highlighted in yellow.

     Sincerely,

       Michael D. Sevilla

Reviewer 2

Review of the submission “Electron-Induced Repair of 2'-Deoxyribose Sugar Radicals in DNA: A Density Functional Theory (DFT) Study” submitted to the International Journal of Molecular Sciences by Bell, Kumar and Sevilla.

This is an interesting and well written article. I have found a few typos and I have a few comments that should be easy to fix.

Comments and Responses  

Comment 1.  Page 3, line 14.  This should be .. in this study is a one electron process.  

 Response:  Corrected as suggested

Comment 2.  Page 3, line 19.  These ΔG’s  are way too big. 

 Response: corrected as suggested

Comment 3.  Page 3, line 20.  I think that should be ….Fermi-Dirac statistics.

 Response: corrected as suggested

Comment 4. Page 4, line 40 .  The prime symbols 1ʹ etc here are way too small. nb there are lots of these in this manuscript. 

 Response- Now changed the superscript/subscript to  normal font size in every where in the revised manuscript.

Comment 5. Page 5, line 75.  (see eq 5)?  This is on page 11 and doesn’t seem right? 

 Response- Eq 5 is correct and accounts for the SHE potential in a straightforward and innovative manner. The usual methods of computing the energy changes for 3 and 5 using the proton solvation energy give the same results within 0.1 V.   We now include a description of both methods in the text and in the supporting information in Table S2.

Comment 6.  Page 6, line 23.    …presents in a 3-dimensional visual plot?  I don’t see this plot in reference 88?

 Response-In Ref 88, we calculated the σ and π radicals of several one-electron oxidized DNA bases and plotted the 3-dimensional spin density which clearly shows that for  π radical the distribution is delocalized over the molecule while for  σ radical the spin density was localized around the radical site.

Comment 7.   Page 6, line 36.  Here again the C1ʹ is way too small.  Same thing on P8 line74.

 Response-Now visible as normal font size in everywhere in the revised manuscript.

Comment 8.   Page 10, line 46.  …using the free energy?  The free energy of what?

Response- We made the following change as suggested: “1.45 eV calculated from the free energies of product and reactant.”

Comment 9.   Page 11, line 91.  “resembles with”?  this should be something like plot of anions resembles that

Response- We made the following change as suggested:  “and the HOMO plots of the anions resembles the spin density plots for the radicals as expected.”

Comment 10.   Page 11, line 95.  “in compare to bases”?  this should be something like …in comparison to the DNA bases”.

Response- We made the following change as suggested:  “Thus, sugar radicals in DNA are far more probable sites to be reduced by the solvated electron in comparison to the DNA bases”

Comment 11.   Page 12, line 23,  make this …involves a substantial barrier.

Response--Corrected as suggested by the reviewer.“electron-induced proton transfer mechanism, the repair of 5',8-cyclo-2'-dG involves a substantial barrier.”

Comment 12.  Page 12, line 41.  Remove .Gamma. and replace with …. γ-irradiated …        Response: Corrected as suggested

Comment 13.  Page 12, line 47.  OF should be of, and And should be and…

 Response: Corrected as suggested

Comment 14.   page 12, line 56.  No authors listed.  So add, Dizdaroglu, Lloyd.  

  Response: Corrected with full reference 11

.

Comment 15.  Page 14, line 55.  Don ‘t need this † here.   

   Response: Corrected as suggested

Comment 16.  Page 14, line 64, H. Bul. should be H·and OH·    

   Response: Corrected as suggested

Comment 17.   Page 15, ref 73,  Ds should be dS, and Ds should be dS. 

    Response: Fixed as ds and ds

Comment 18.  Page 15,  ref 77 this should be pKas  

    Response:  Fixed as suggested

Round 2

Reviewer 1 Report

I find the corrected version suitable for publication.